# An equitable redistribution of unburnable carbon

Steve Pye [1✉], Siân Bradley[2], Nick Hughes[3], James Price[1], Daniel Welsby [3] & Paul Ekins[3]

The rapid phase-out of fossil fuels is critical to achieving a well-below 2 °C world. An emerging body of research explores the implications of this phase-out for fossil fuel producing countries, including the perceived tension between least-cost and most-equitable pathways. Here we present modelling, which re-distributes remaining fossil fuel production towards developing countries. We show that redistribution is challenging due to large economic disincentives required to shift production, and offers limited economic benefit for developing countries given the long timeframe required to effect change, and the wider impact of rising fuel import and energy systems costs. Furthermore, increases in production shares are offset by shrinking markets for fossil fuels, which are part dependent on carbon capture and storage (CCS). We argue that while there is a weak economic case for redistribution, there is a clear role for equity principles in guiding the development of supply side policy and in development assistance.

[1] UCL Energy Institute, University College London, Central House, 14 Upper Woburn Place, London WC1H 0NN, UK. [2] The Royal Institute of International Affairs, Chatham House, 10 St James's Square, London SW1Y 4LE, UK. [3] UCL Institute for Sustainable Resources, University College London, Central House, 14 Upper Woburn Place, London WC1H 0NN, UK. ✉email: s.pye@ucl.ac.uk

To remain within a well-below 2 °C carbon budget, rapid reductions in the production and use of fossil fuels are required. Scenarios presented in the IPCC special report on 1.5 °C suggest a decline of coal to 18% of 2020 levels in 2050, oil to 34% and natural gas to 57% (median values)[1]. These and almost all 2 °C scenarios have profound implications for future fossil fuel production, and for producer countries facing the prospect of fossil fuel reserves being stranded or left undeveloped. In their 2015 paper, McGlade and Ekins[2] analysed global fossil fuel production under a 2 °C scenario, highlighting that a cost-optimal allocation leads to high level of unburnable reserves, with clear winners and losers across different regions.

One criticism of the cost-optimal approach, which is inherent in the structure of Integrated Assessment Models[3], is that it ignores the political economy of fossil fuel production and use, including equity considerations such as which countries get to produce their fossil fuel reserves as global production declines. This is problematic from a developing country perspective, where calls to keep it in the ground have been questioned given the historical benefits that many developed countries have realised from their fossil fuel reserves[4].

Questions of equity in climate policy have long been part of the UNFCCC process. The Paris Agreement recognised the long-held principle of common but differentiated responsibilities, and established a framework under which countries pledge Nationally Determined Contributions (NDCs) to global emissions reductions targets, which reflect their differing circumstances and capacities[5]. These national level targets apply to emissions arising at the point of fossil fuel consumption. Accordingly, equity has been largely addressed in terms of consumption, where national limits on emissions would limit the domestic consumption of fossil fuels, which might in turn constrain economic growth in developing countries. By contrast, the issue of equity in production has historically been avoided, in part due to the risk of producer country claims for loss and damage, which could severely undermine climate negotiations and finance.

There is now growing interest in the contribution that supply side policies, which aim to curb fossil fuels at the point of production, can make to fossil fuel phase-out[6,7]. Lazarus and van Asselt set out the range of supply-side policies, including economic instruments such as subsidy removal, the taxation of fossil fuel production and the development of tradable production allowances, and regulatory approaches including calls to restrict exploration[8]. Piggot et al. note emerging momentum around supply-side policy at the international level, and argue for a more explicit recognition of the need for action on fossil fuel production under the UNFCCC process[9]. Asheim et al. argue that a supply-side treaty could complement demand-side action under the Paris Agreement and help coalesce action amongst producer countries[10].

These developments have clear equity implications. Caney[11] and Kartha et al.[12] argue that policies that result in the stranding of fossil resources for some countries and not others will inescapably give rise to issues of equity. Kartha et al. argue that leaving the question of who may extract to markets risks leaving the least-able carrying the greatest burden, and that countries cannot be expected to constrain fossil fuel production unless this is part of a broader effort by the international community to reduce production[13]. Common to each is a focus on the perceived opportunity cost incurred by developing countries that forgo fossil fuel production, particularly the revenues, fuel flows and employment that fossil fuel development can in theory provide.

There is relative agreement that serious consideration of equity would lead to higher-income and higher-capacity countries constraining their production first, in line with both the reductions needed to meet the objectives of the Paris Agreement, and the principle of common but differentiated responsibility. For example, the Lofoten Declaration on the Managed Decline of Fossil Fuels states that developed countries have a responsibility and moral obligation to take the lead[14]. Kartha et al. argue that an equitable approach would minimise economic disruption, supporting economic diversification and ensuring the provision of energy services and investment in job creation, and be based on a fair distribution of costs[13]. Muttitt and Kartha go on to set out a set of principles to apply to an equitable transition away from fossil fuel production[15].

Three broad challenges arise when thinking about how an equitable decline in production might be approached. The first is how to define an equitable distribution between countries. Caney proposes three criteria; a country's current stage of development, its ability to develop based on non-fossil fuel alternatives, and its historical responsibility based on previous production and associated benefits accrued[11]. Second is whether and how to account for potential production that is foregone or reduced. Placing a monetary value against undeveloped resources is problematic, with large uncertainty around future production and potential revenues. Third is the presumed tension between equitable and cost-optimal (economic efficiency) approaches. Lenferna argues that prioritising equity criteria could result in high-cost reserves and those that require new infrastructure being developed, while low-cost reserves and those with infrastructure in place are phased out. A focus on reserves where equity and efficiency incentives overlap could be a focus, e.g. prioritising the stranding of inefficient reserves in rich countries, for example Canada's oil sands or Norway's high north region[16].

The supply-side literature and the equity arguments therein have to date not included any quantitative analysis of the implications of an equitable approach to managed decline, including the tension between equity considerations and economic efficiency. Building upon analysis in McGlade and Ekins[2], this modelling with TIAM-UCL contrasts a cost-optimal distribution of fossil production with an equitable redistribution, which incorporates equity considerations using two approaches as outlined above; first, and the focus of this paper, related to a country's level of development as measured by the Human Development Index (HDI), and second, based on a country's accrued benefit from past production.

We find that very large economic disincentives are required to move production to low-medium human development (LMHD) regions, and that LMHD production increases only in later decades, when global demand for fossil fuels is lower. Redistribution also raises the costs to the energy system, increasing costs on non-producer importing countries, suggesting a tension between equity in production and equity in consumption. While we conclude that the case for an HDI-based equitable redistribution may be overstated and could disadvantage LMHD countries that are import dependent, we argue that meaningful engagement with equity principles can help inform supply-side policy in the highest human development (VHHD) countries and ensure the recognition of LMHD perspectives in international cooperation on fossil fuel supply.

## Results

**Global energy system modelling**. The modelling, described in the Methods section, is undertaken using the global energy system model, TIAM-UCL[17–20]. The mechanism to redistribute production based on the above criteria is a carbon tax on fossil fuel production, differentiated by region. Concerning development need, defined using the HDI, very high human development (VHHD) countries are subject to a high carbon-based production tax, high human development (HHD) countries to a lower tax,

while LMHD countries are exempted (Methods). In the accrued benefit approach, similar groupings are created based on historic fossil fuel rents (Supplementary Note 3). The model is first run allowing for a cost-optimal allocation under climate targets, and then subsequently run with the addition of the redistributive mechanism (carbon taxes on production) in place. The scenarios are assessed under high and low tax variants under 1.75 and 2 °C warming objectives. Note that we do not propose this redistributive mechanism as a basis for international policy, but as a modelling mechanism with which to explore the impacts of an equitable redistribution. In doing so, we explore the implications of changing distribution both for producers and consumers of fossil fuels. The complexities of operationalising an equity approach are further considered later in this paper.

The modelling results provide three important insights. First, under the global climate ambition pursued, any redistribution of fossil fuel production will be in a declining market, with this decline more rapid as climate policy stringency increases. The result is that smaller, less established producers from LMHD regions will be competing with large incumbent producers as market size reduces and prices fall. Therefore, the benefits of redistribution will be limited. The 1.75 °C pathways see reductions of up to 59–62, 59–62 and 87% for gas, oil and coal, respectively, in 2060, relative to current levels (Fig. 1c, i for gas and oil, respectively; Supplementary Fig. 1 for coal). For 2 °C, the comparable figures are 48–54%, 51–60% and 75–80% (Fig. 1f, l for gas and oil, respectively; Supplementary Fig. 2 for coal). Given the large reduction in coal and limited redistribution potential, the focus of the redistributed scenarios is on oil and gas under the 1.75 °C target, consistent with the ambition of the Paris Agreement.

What market does remain is itself subject to critical uncertainties. Demand growth may not be as robust as suggested under the narrative used, namely SSP2 (see Methods). A sensitivity case using lower energy demands (based on SSP1) suggests lower levels of fossil fuel production, and lower shares for LMHD, particularly for oil (Supplementary Note 2). Indeed, the continued production of oil and gas as shown in these scenarios is contingent upon rapid declines in coal consumption, which look optimistic given recent trends, and by the availability of bioenergy with carbon capture and storage (BECCS) in the system. For example, in the 1.75 °C cases, negative emissions from BECCS offset 2.7–3.7 GtCO$_2$ in 2060, sufficient to capture 80–90% of the CO$_2$ emissions from the gas produced (and used unabated) or 40–60% from oil.

Second, large disincentives are required to redistribute production towards LMHD regions (Africa, Other Developing Asia (ODA), India) and away from established HHD (Russia and Middle East) and VHHD (USA) producers, even in a shrinking market. This is also reflected in the accrued benefit scenarios (although the different criterion affects regions differently, with for example the Middle East and Former Soviet Union being included in the high benefit group, therefore incurring the highest tax rate (see Supplementary Note 3)) and highlights the challenge of moving away from allocating resources on the basis of cost-effectiveness. In the 1.75 °C low and high tax scenarios, for gas, significant production share gains for LMHD regions are only observed after 2040 (Fig. 1a, b), taking an additional 38/57% share (in 2050/60) mainly at the expense of HHD regions. This requires tax rates on HHD production of $188–375/tCO$_2$ in 2050 and thereafter. Relative to the cost-optimal case, gas production levels in 2040 in LMHD are about the same, but then are 2–3 times higher in 2060 (Fig. 1c). Under the 2 °C scenarios, the larger market size allows for marginally higher increases in production in LMHD regions, compared with under 1.75 °C. The increase in share, however, is lower, with HHD regions retaining higher market share (Fig. 1e, f).

For oil, in the 1.75 °C low and high tax scenarios, the production taxes mean that LMHD regions increase market share by 2040 relative to the cost-optimal case (Fig. 1g, h), but only at similar (not increases on) production levels observed in 2020, again at the expense of producers in HHD regions (Fig. 1i). In the 2 °C scenarios, a similar pattern is observed, although the market share for LMHD group is lower, as in the case for gas, due to the higher global production levels and HHD share (Fig. 1j–l).

Third, moving away from a cost-optimal approach and toward an equitable redistribution increases the overall costs of the system. The trend lines in Fig. 2a–d show the additional costs incurred by HHD and VHHD regions strongly outweigh the benefits to LMHD regions. The same levels of global fossil fuel production are more costly due to higher-cost fossil resources being extracted, requiring additional infrastructure investment and incurring operational costs (see Supplementary Fig. 3). This reveals a tension between different equity perspectives; the benefits for LMHD producers come at a cost for the consumers in importer countries, including in the LMHD group. Furthermore, cost savings are unequally distributed across LMHD regions, and are only realized at scale later in the period, when tax disincentives are much higher. It is also possible that the higher production costs arising from a redistribution could harm those intended most to benefit. This outcome is seen in the accrued benefit 1.75 °C cases, in which the low benefit group which increases production, actually increases its domestic system costs as much of their higher-cost fossil fuel is consumed domestically and not exported (Supplementary Fig. 14).

Figure 3a–d details the balance of commodity trade costs, a key component of the energy system cost. A reduction is observed for the LHMD regions, as their fossil fuel exports increase and import dependency reduces, with the opposite true for HHD and VHHD regions, except in the near terms for the HHD group under the 2 °C cases. The benefits for LMHD regions shown here are much higher than those shown in Fig. 2, as they are not net of the investment and running costs associated with production, often for relatively higher-cost resources. The relative share of fuel trade savings versus the higher costs of production are illustrated in Supplementary Fig. 3.

## Discussion

We find that the potential benefits of an equitable redistribution of future fossil fuel supply, which provides developing countries greater space for fossil fuel production within a given carbon budget, may be overstated. The evidence presented here—from the strength of disincentives and the timeframe required to re-allocate production toward LMHD countries, to the uncertainties around the future size of markets, to the global increase in trade and energy systems costs and their unintended consequences for energy importers, some of which are LMHD countries—suggests that an equitable redistribution may not bring the level of benefits perceived, and could disadvantage LMHD countries that rely on imports. The ethical basis and practical criteria for operationalising a redistribution of producer rights would present challenges. Our illustrative modelling of two such approaches—based on human development need and historic benefit—suggests that either could create controversies or unintended consequences. This raises questions about how helpful consideration of the equitable redistribution of production is.

The assumption of benefits foregone, which underpins much of the literature on equity considerations, also warrants scrutiny. Supply-side policies such as the World Bank's decision to stop financing upstream oil and gas[21] and the UK Environmental Audit Committee's recommendation to halt export credit for fossil fuels in developing countries have been questioned on the

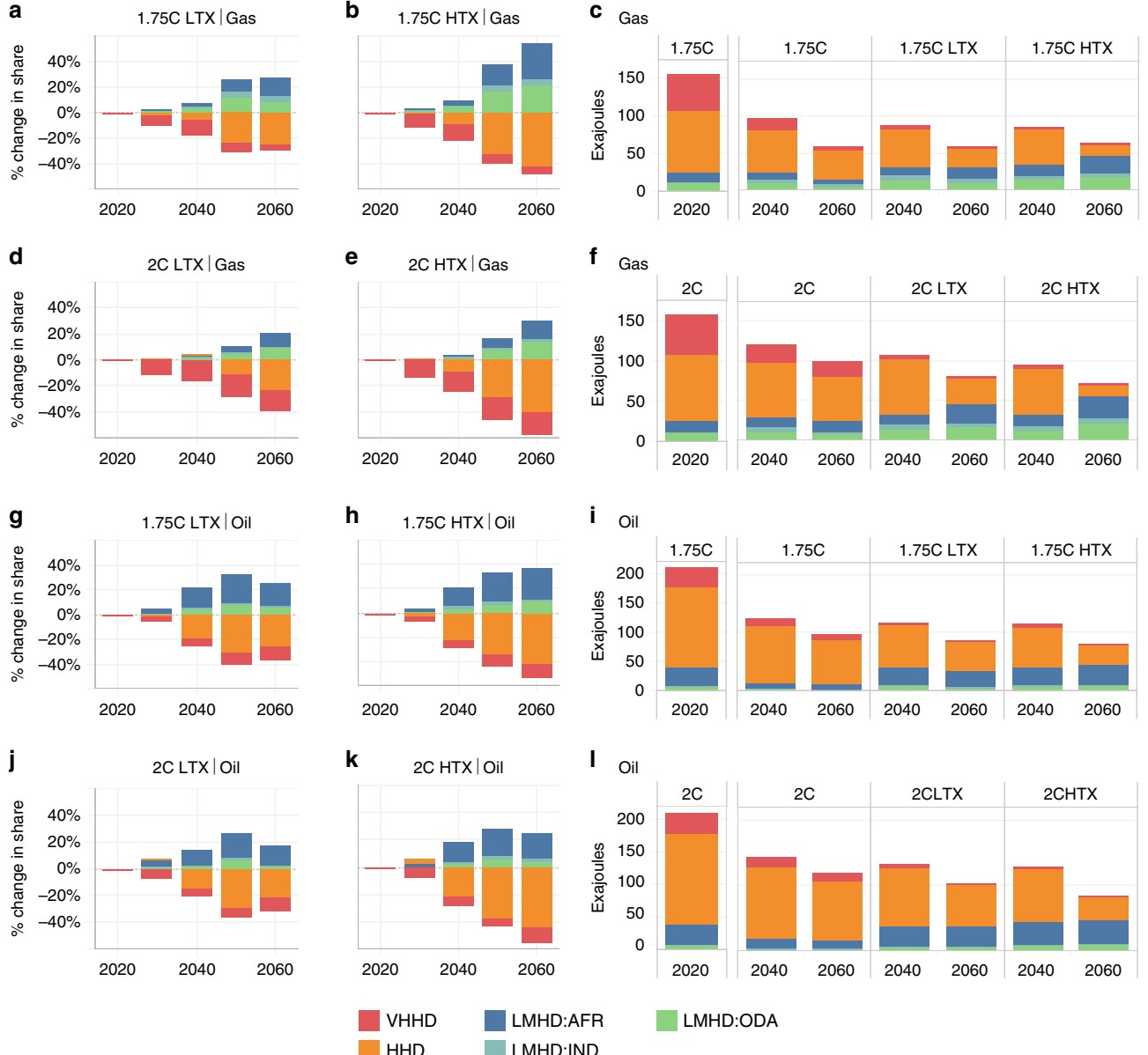

**Fig. 1 Change in production levels of fossil fuels under 1.75 and 2 °C redistributed cases relative to the cost-optimal case, 2020–2060.** The left hand panel shows the annual % change in production for a region relative to total global production under the cost-optimal case. Panels (**a**, **b**) and (**d**, **e**) show the change in gas production under low (LTX) and high tax (HTX) levels for 1.75 °C and 2 °C cases, respectively. Panels (**g**, **h**) and (**j**, **k**) show the same information for oil production. A positive change reflects a gain in total production in the given region, where, for example, 20% reflects an increase share equivalent to 20% of global production in the cost-optimal case. The right hand panel shows the absolute production levels, necessary for putting the percentage changes (left hand panel) in context. Panels (**c**) and (**f**) show the gas production level under low and high tax levels for 1.75 and 2 °C cases, respectively. Panels (**i**) and (**l**) show the same information for oil production. See Table 1 for the allocation of regions to the three HDI groups: VHHD, HHD, and LMHD. Source data are provided as a Source data file.

basis of equity and just transition[22]. However, it is evident that not all countries with fossil fuel reserves have achieved high levels of development, showing that the translation of fossil fuel flows and revenues into sustainable development is not straightforward or guaranteed. Furthermore, the development of the infra-structures and markets required to utilize domestic production tends to lock-in rising fossil fuel consumption.

Where access to energy is concerned, the benefits of lower commodity trade costs for LMHD regions are questionable when considered against the declining costs of renewable alternatives, which already undercut fossil fuels in many markets and will do globally by the mid-2020s[23]. In this context, an equitable

redistribution may increase the exposure of LMHD regions to carbon risks and delay their transition[24].

Where the economic benefits of fossil fuel production are concerned, these should be viewed in light of the partnerships that are required for exploration and development. Developing countries with low levels of domestic financial and technical capacity are typically reliant on foreign investors and operators in order to monetise their resources. Any benefits of an equitable redistribution of fossil fuel production would not only be con-ferred to producer countries, but also to wealthy international companies. Fiscal regimes for oil and gas production also tend to allow operating companies to write-off costs against initial

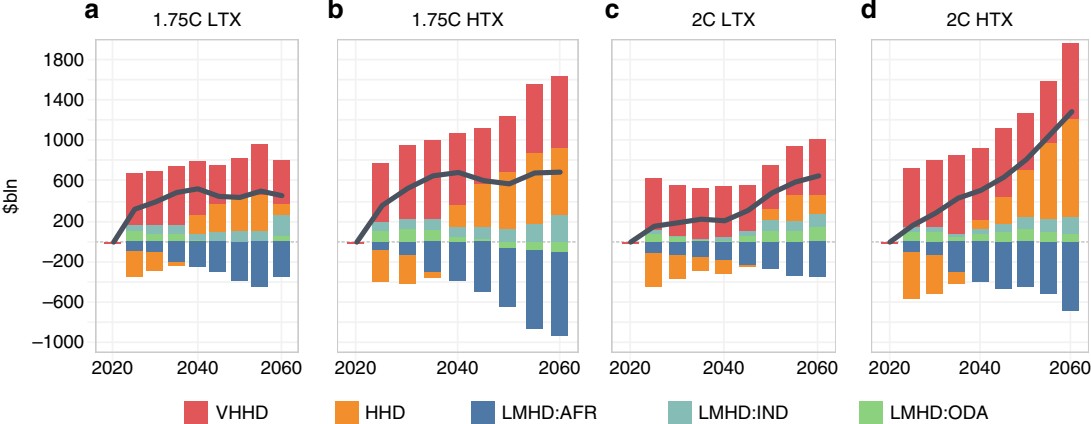

**Fig. 2 Change in regional energy system costs and commodity trade costs under 1.75 and 2 °C HDI-based redistributed cases relative to optimal cases, 2020–2060.** Panels **a–d** represent the change in energy system costs by HDI group, compared with the cost-optimal case. Negative values show a reduction in costs, while the black trend line shows the net change in global costs. Cost estimates do not include tax revenues raised by the production tax, with the focus on the techno-economic costs of the system. Source data are provided as a Source data file.

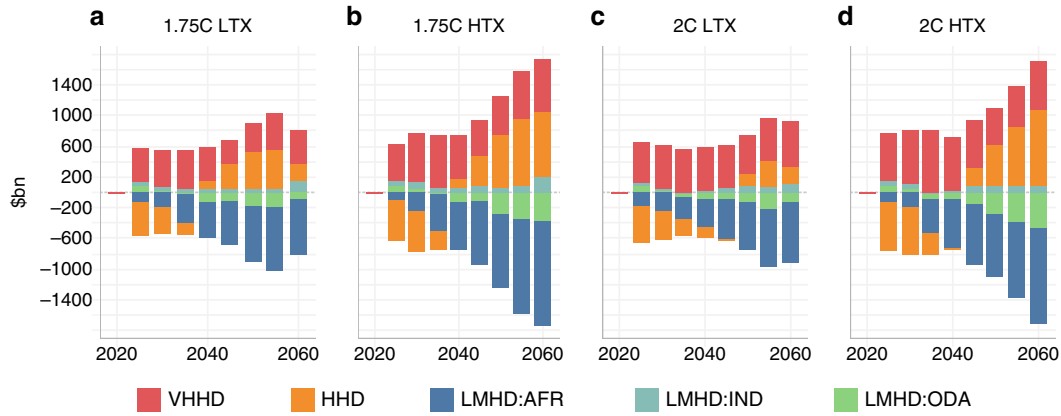

**Fig. 3 Change in commodity trade costs under 1.75 and 2 °C HDI-based redistributed cases relative to optimal cases, 2020–2060.** Panels **a–d** show the change in commodity trade costs by HDI group, compared with the cost-optimal case, reflecting the balance between export revenues and import costs. The net level is zero in all cases. Negative values show an increase in net exports. Trade cost estimates do not include tax revenues raised by the production tax, with the focus on the techno-economic costs of the system. Source data are provided as a Source data file.

returns, before the host country sees any meaningful income from the sector. As the timeframes for viable fossil fuel production tighten, the prospect that LMHD regions see limited economic returns will increase.

Equity considerations do still have a crucial role to play in guiding the managed decline of fossil fuels. The decision to produce fossil fuels is ultimately a sovereign one, and as with consumption-focused emission reduction targets, there is little prospect of top-down binding obligations at present. However, this does not preclude stronger recognition of supply-side policy measures and commitments, particularly given that the fossil fuel sector and associated industries are typically among the largest drivers of energy demand and emissions in-country. Better understanding of what an equitable distribution might look like between countries could help inform unilateral or collective supply-side commitments that are in line with the voluntary, bottom-up pledge, review and ratchet mechanism of the Paris Agreement, and which acknowledge the primacy of domestic policy and the need to avoid distributional conflicts[25].

Meaningful engagement with equity considerations in the context of managed decline should continue to highlight the need for advanced economies to demonstrate international leadership on the phase-out of fossil fuels. While we argue that the modelling supports the case for a least-cost pathway to managed decline, production subsidies and other market distortions effectively move production away from a cost-optimal distribution, often in favour of more developed producer countries. The development of supply-side policies that are in line with national climate commitments, addressing production subsidies, investment decisions on new fossil fuel supply, and the overall timeframe for managed decline, could help demonstrate a commitment to fairness. It would also help level the playing field for producer countries, ensuring that the least-cost reserves meet remaining global demand for fossil fuels.

Developing country perspectives on managed decline and on alternative pathways to access to energy and economic development must also be acknowledged. Freeing up inefficient fossil fuel finance also raises the prospect of greater financial flows into low-carbon energy systems, including the transfer of finance and technologies to developing countries. While there are few immediate replacements for the high economic rents that oil and gas can generate, there are no shortage of clean energy alternatives. Where production is established, development assistance should help accelerate economic diversification and domestic energy transition, and upstream emissions mitigation. Where production is under consideration, development assistance can make climate-related risks clear and coordinate alternative options for access to energy and economic development[24].

**Table 1 HDI criteria groups and their composition based on regions in TIAM-UCL.**

| HDI group name | HDI group | HDI level | TIAM-UCL regions | Tax level group |
|---|---|---|---|---|
| LMHD | Low-medium human development | <0.7 | Africa, India, Other Developing Asia | 0 |
| HHD | High human development | 0.7–0.8 | Middle East[a], Mexico, South and Central America, China, Former Soviet Union[b] | 1 |
| VHHD | Very high human development | >0.8 | Western Europe, Eastern Europe[c], UK, Canada, USA, Australia, Japan, South Korea | 2 |

[a]The Middle East group includes oil rich Gulf States who have an HDI level over 0.8.
[b]Russia is at 0.798, at the upper end of the HHD range.
[c]The exceptions in this region are Bulgaria and Romania, although they have high scores in the HHD group.

## Methods

**TIAM-UCL model**. To explore the equity issues associated with fossil fuel production under different climate targets, we used the TIMES Integrated Assessment Model at University College London (TIAM-UCL). This model provides a representation of the global energy system, capturing primary energy sources (oil, gas, coal, nuclear, biomass and renewables) from production through to their conversion (electricity production, hydrogen and biofuel production, oil refining), their transport and distribution, and their eventual use to meet energy demands across a range of economic sectors. Using a scenario-based approach, the evolution of the system over time to meet future energy service demands can be simulated, driven by a least-cost objective. The model uses the TIMES modelling framework, which is described in detail in Supplementary Note 5.

The model has a 16 region representation (Table 1), allowing for more detailed characterization of regional energy sectors, and the trade flows between regions. Upstream sectors within regions that contain members of OPEC are modelled separately, so as an example, the upstream sector in the Central and South America (CSA) region will be split into two, with the upstream sector for OPEC including Venezuela and Ecuador. Regional coal, oil and gas prices are generated within the model. These incorporate the marginal cost of production, scarcity rents (e.g. the benefit foregone by using a resource now as opposed to in the future, assuming discount rates), rents arising from other imposed constraints (e.g. depletion rates), and transportation costs but not fiscal regimes.

A key strength of TIAM-UCL is the characterization of the regional fossil resource base (Supplementary Note 6). For oil reserves and resources, these are categorized into current conventional proved and probable (2P) reserves in fields that are in production or are scheduled to be developed, reserve growth, undiscovered oil, Arctic oil, light tight oil, natural gas liquids, natural bitumen, extra-heavy oil and kerogen oil. The latter three categories represent unconventional oil resources. For gas, these resources are categorized into current conventional 2P reserves that are in fields in production or are scheduled to be developed, reserve growth, undiscovered gas, Arctic gas, associated gas, tight gas, coal-bed methane and shale gas. Categorisation of resources and associated definitions are comprehensively described in Chapters 2 and 3 of McGlade[26]. For oil and gas, individual supply cost curves for each of the categories are estimated for each region. Crucially, the upstream emissions associated with the extraction of different fossil fuels is also captured in the model.

The model does have the ability to remove emissions from the atmosphere via negative emissions, based on a set of Bioenergy with carbon capture and storage (BECCS) technologies, in power generation, industry, and in $H_2$ and biofuel production. The primary limiting factor on this technology is the global bioenergy resource potential, set at a maximum 110 EJ per year, in line with the recent UK Committee on Climate Change (CCC) biomass report[27]. This is a lower level than the biomass resource available in many other integrated assessment scenarios for 1.5 °C (which can be up to 400 EJ/yr)[28,29], and is more representative of an upper estimate of the global resource of truly low-carbon sustainable biomass based on many ecological-based studies[30] (Supplementary Table 13).

In the model, future demands for energy services (including mobility, lighting, residential and industrial heat and cooling) drive the evolution of the system so that the energy system in 2050 meets the energy services required, which have increased through the population and economic growth. For this paper, we use Shared Socio-economic Pathway 2 (SSP2) derived energy service demands[31], plus undertook sensitivity cases using SSP1 demands (Supplementary Note 2). Decisions around what energy sector investments to make across regions are determined on the basis of the most cost-effective investments, taking into account the existing system to 2017, energy resource potential, technology availability and crucially policy constraints such as emissions reduction targets. The model time horizon runs to 2100, in line with the timescale typically used for climate stabilization.

A climate module is also integrated into the model framework, calibrated to the MAGICC simple climate model used by the IPCC[32], allowing for a simplified representation of the climate system. It ensures that any future energy system is consistent with a given temperature objective, such as limiting warming to 1.75 or 2 °C by 2100 and beyond. To this end, the climate module is run out well beyond 2100 to ensure emission levels in 2100, and the underlying energy system, are consistent with maintaining a stable temperature from that point onwards. As described later,

this approach is coupled with carbon budgets in order to be more representative of the most up-to-date climate science assessed by the IPCC that used multiple lines of evidence to estimate carbon budgets and not just simple climate models.

Further information on key assumptions used in the model is provided in Supplementary Note 7. The TIAM-UCL model version used for this analysis was 4.1.1, and was run using TIMES code 4.3.4 with GAMS 24.7. The model solver used was CPLEX 12.6.3.0.

**Defining climate ambition for scenarios**. The purpose of modelling an equitable scenario is to explore the implications of a redistribution mechanism for production levels and costs, relative to a cost-optimal perspective. The modelling is illustrative, in the sense that it is not intended as a direct simulation of a particular portfolio of policies, but as an exploration of the implications of redistribution, in the absence of quantitative analysis in this area to date. The approach to the analysis is based on the comparison between two types of model runs under two different levels of climate ambition:

- (i) Cost-optimal allocation of production within a 1.75 and 2 °C carbon budgets.
- (ii) Equitable-redistribution cases, simulated by introducing a differentiated carbon tax on production based on development level and accrued historic benefits.

Differing levels of climate ambition are modelled by implementing a range of global carbon budgets within the model, the assumptions for which are based on the latest science, and taken from the IPCC SR1.5 report, Table 2.2[1]. Two levels of climate ambition have been modelled, for 1.75 °C and 2 °C temperature targets, at 66% probability, using carbon budgets from 2018 onwards of 800 $GtCO_2$ and 1170 $GtCO_2$, respectively. The climate module is also used to ensure that warming does not exceed 2 °C under any scenario, and that temperature targets are hit in 2100. Overshoot in the 1.75 °C case is permitted due to the inability of the model to remain below these limits due to a combination of a warming in 2005 of 0.86 °C (taken from Table 1.1 of the SR1.5 report) and strong energy demand growth in the near term.

Budgets for 1.5 °C temperature targets at 66% probability (420 $GtCO_2$) and 50% probability (580 $GtCO_2$) have not been run in this analysis, but are further explored in a report to the UK Committee on Climate Change[33]. That analysis reported that, compared with other published 1.5 °C scenarios[28], TIAM-UCL did not produce a solution that met this objective due to a number of factors, including an assumed lower global biomass potential (to reflect sustainability limits), lower CCS deployment rates, and higher residual emissions in industry and transport. However, the analysis also showed that this 1.5 °C ambition could be achieved under lower demand growth and with greater use of nature-based $CO_2$ sequestration options (e.g. afforestation).

For $CH_4$ and $N_2O$ emission levels, non-energy sector related emissions (outside of the energy system) are fixed based on a RCP2.6 trajectory. This sees non-$CO_2$ emissions decline rapidly (particularly for methane emissions) over the next few decades and, when coupled with sufficiently rapid reductions in $CO_2$ emissions, would be approximately consistent with keeping median expected peak warming around 1.5 °C. However, emissions of non-$CO_2$ GHGs from the energy system are endogenously modelled, with mitigation options available, for example, to reduce $CH_4$ leakage from oil and gas extraction and supply activities. This allows the model to replicate the necessary action of increasing the stringency of environmental regulation for oil and gas systems. Without action to address such emissions, it is likely that fossil fuels would need to be phased out more rapidly. The sum of the exogenous and endogenous non-$CO_2$ GHGs emissions are required (via a constraint) to be below a declining emission trajectory out to 2100. For $CH_4$ and $N_2O$, these constraints are derived separately and are based on the mean of pathways used in the IPCC Special Report on 1.5 °C (here after SR1.5) that meet a 2 °C climate target in 2100[28] (noting that this includes scenarios from the SR1.5 database that achieve close (+/−0.1 °C) to the 2 °C target in 2100).

**Modelling a redistribution mechanism**. The redistribution mechanism chosen to simulate equity principles is a carbon tax on production. Two sets of equity criteria

have been tested; firstly, current level of development, and secondly, accrued benefits from past production. The primary focus of the paper is on the level of development, as measured by the Human Development Index (HDI)[34]. This is because the research is motivated by debates within the climate and development community regarding the potential for development based on fossil fuel extraction versus alternative low-carbon pathways, and the responsibility of high income and high capacity fossil fuel producing countries to take the lead in respect of mitigation, and support countries with lower level of development and less capacity. An alternative criteria applied for comparison concerns historical benefit from fossil fuel production. The rationale for this perspective is that countries that have gained the most historically should provide opportunities for countries who have benefited

less to increase their production going forward. Further discussion on the use of these criteria is provided in Supplementary Note 4.

Concerning the core equity criterion, HDI thresholds used to categorise model regions into three groups are shown in Table 1. For model regions that are single countries, such as China, this was straightforward. For model regions that contain multiple countries (Supplementary Table 18), we determined a population-weighted mean HDI score, based on the individual country scores and aggregated to the region level. As a result, there are some countries with very different index scores to the group in which their region is allocated, notably the Middle East region. Oil rich Gulf states, which have index scores >0.8 (VHHD) are included in the model's Middle East region and therefore the HHD group.

The accrued benefits from production criterion is based on cumulative rents between 1970 and 2017, based on World Bank data[1], and provides an alternative distributive basis for comparison with the HDI criterion. The approach, described in Supplementary Note 3, categorises countries into low, medium and high benefit groups, and reflects the resource gains as measured by rents of different countries, on a per capita basis to account for population. However, this does not imply that these accrued benefits have necessarily led to higher development or been equitably spread.

Having used different equity criteria to establish regional groupings, a carbon tax mechanism was used to simulate a redistribution of regional production away from cost-optimality. This mechanism is not being proposed as a policy instrument, but as a modelling device to set up an illustrative equity-based scenario. The carbon tax is specifically added to fossil fuel production for oil, gas and coal, based on the carbon content of different fossil fuels, and therefore the same carbon tax level results in relatively higher additional cost for coal, compared with oil and gas. In addition, it will also result in higher (per unit of production) costs for more energy intensive technologies which produce the same energy commodity, such as the extraction of synthetic crude oil from tar sands. The tax does not apply to the wider emissions associated with extraction, for example on processing and transportation (although these emissions are included in the model), with the exception being fugitive $CH_4$ emissions from production of natural gas. This is due to the importance of this issue on the role of natural gas in the transition[35–37].

Two tax trajectories shown in Fig. 4 have been derived for the groups shown in Tables 1 and 2. The Group 2 trajectories (blue trend line) are applied to production in those regions that have the highest development level (VHHD), or for the alternative criteria, have benefited the most from previous production (HBN). A high (continuous line) and low (dashed line) variant are used to explore production levels under different tax levels. The Group 2 trajectories start at $100/tCO₂,

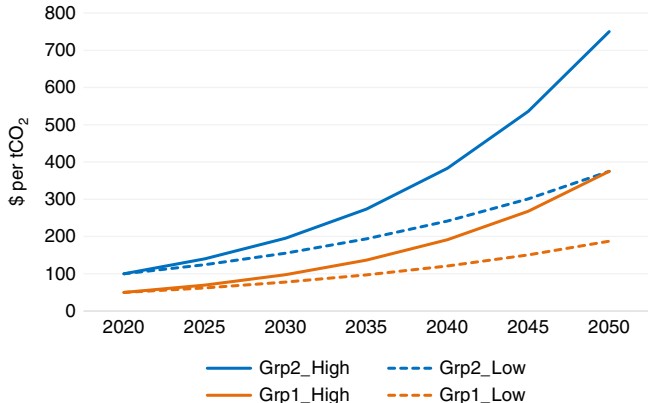

**Fig. 4 Carbon tax levels applied to fossil production in different HDI groups, 2020–2100. a** Different carbon tax level trajectories applied across different HDI groups, with upper side of wedge being high tax variant and lower side of wedge being low variant. **b** Tax level trajectory variant for more rapid increase in tax levels, where high tax level in 2100 under (**a**) is implemented by 2050, and then held constant to 2100. Source data are provided as a Source data file.

**Table 2 Accrued benefits criteria groups and their composition based on regions in TIAM-UCL.**

| Benefit group name | Benefit group | Accrued benefit level ($ per cap.) | TIAM-UCL regions | Tax level group |
|---|---|---|---|---|
| LBN | Low | <5000 | India, Other Developing Asia, China, Japan, South Korea, Eastern Europe | 0 |
| MBN | Medium | 5000–20,000 | Africa, Mexico, South and Central America, Western Europe, UK | 1 |
| HBN | High | >20,000 | Canada, USA, Australia, Middle East, Former Soviet Union | 2 |

**Table 3 List of scenarios modelled in TIAM-UCL.**

| Scenario type | Scenario name | Description |
|---|---|---|
| Optimal | 1.75C | Optimal distribution of fossil fuel production under 1.75 °C target. Global carbon budget of 800 GtCO₂ from 2018, with warming limited to 1.75 °C by 2100. No overshoot of 2 °C permitted. |
| | 2C | Optimal distribution of fossil fuel production under 2 °C target. Global carbon budget of 1170 GtCO₂ from 2018, with warming limited to 2 °C by 2100. No overshoot of 2 °C permitted. |
| Redistributive[a] | 1.75C HDI LTX | HDI-based redistributive case under a 1.75 °C target using a lower tax variant |
| | 1.75C HDI HTX | HDI-based redistributive case under a 1.75 °C target using a higher tax variant |
| | 2C HDI LTX | HDI-based redistributive case under a 2 °C target using a lower tax variant |
| | 2C HDI HTX | HDI-based redistributive case under a 2 °C target using a higher tax variant |
| | 1.75C-A HDI LTX | HDI-based redistributive case under a 1.75 °C target under SSP1 demands, using a lower tax variant |
| | 1.75C-A HDI HTX | HDI-based redistributive case under a 1.75 °C target under SSP1 demands, using a higher tax variant |
| | 2C-A HDI LTX | HDI-based redistributive case under a 2 °C target under SSP1 demands, using a lower tax variant |
| | 2C-A HDI HTX | HDI-based redistributive case under a 2 °C target under SSP1 demands, using a higher tax variant |
| | 1.75C BEN LTX | Accrued benefit-based redistributive case under a 1.75 °C target using a lower tax variant |
| | 1.75C BEN HTX | Accrued benefit-based redistributive case under a 1.75 °C target using a higher tax variant |
| | 2C BEN LTX | Accrued benefit-based redistributive case under a 2 °C target using a lower tax variant |
| | 2C BEN HTX | Accrued benefit-based redistributive case under a 2 °C target using a higher tax variant |

[a]Higher tax variants in 2030/50 are $196/$750 (Group 2) and $98/$375 (Group 1), while for lower tax variants, the 2030/50 levels are $155/$375 (Group 2) and $78/$188 (Group 1).

increasing by 7 and 5% per annum, resulting in 2050 levels of $750 and $375. These levels are maintained to the end of the model horizon (2100). The starting tax level is reflective of some country carbon tax levels in place today[38]. For Group 1 countries, the tax trajectories (orange trend lines) are at 50% of the level for Group 2. This level was determined based on the distribution of population-weighted HDI scores across the three groups, where the Group 1 score of 0.74 was in the middle of the range between Group 2 (0.89) and 0 (0.58). Group 0 countries do not incur any carbon tax on production. Both equity cases followed the same basis for comparability (Table 3).

**Reporting summary**. Further information on research design is available in the Nature Research Reporting Summary linked to this article.

## Data availability

The results data and key source data (shown in the Supplementary Figures) are provided with this paper. Other datasets used in the determination of the equity criteria include the Human Development Index (accessed here http://hdr.undp.org/en/data) and fossil fuel rents data from the World Bank World Development Indicator set (accessed here https://databank.worldbank.org/). Other modelling input assumptions are available on reasonable request. Source data are provided with this paper.

## Code availability

The code underlying the TIAM-UCL model is available at this link https://github.com/etsap-TIMES/TIMES_model. Source data are provided with this paper.

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

## Acknowledgements

This study was conceived of under a programme of work supported by the UK Department for International Development, and subsequently developed. We would like to acknowledge the input of Glada Lahn from Chatham House in helping to develop the ideas formulated in this paper.

## Author contributions

S.P. and P.E. conceived of the research idea. S.P. and S.B. developed the methodology and framing. S.P. and J.P. undertook the energy modelling, and analysed the results. S.P. and D.W. developed the supplementary information. S.P. and N.H. developed early drafts of the paper. All authors contributed to writing the final paper.

## Competing interests

The authors declare no competing interests.
