## [Peer Review File · Nature Communications]

Reviewers' comments:

Reviewer #3 (Remarks to the Author):

This article examines the distributive implications of fossil-producing countries sharing the depleting market for fossils based on economic efficiency or equity in stringent climate stabilization scenarios. Past literature on equity has been dominated by mitigation cost-sharing, until recently. The authors correctly point out that there has been growing attention to the issue of stranded fossil assets, including for developing economies that rely on fossils for supply and export. This article would be the first to quantitatively examine the financial implications of equity frameworks in distributing future fossil production. In this sense, the article is welcome.

However, this article picks up on only one aspect of an equitable view of fossil extraction (how to allocate the remaining fossil production 'budget' – there are many others, as pointed out by the authors in their introduction (e.g. employment, diversification away from fossil dependence, stranded costs, etc). For a high-impact journal, the authors need to do a far more thorough analysis of this one aspect if they are to provide robust insights for policy. The authors choose a very limited set of scenarios from a much wider range of possibilities (as described below). A deeper exploration would potentially provide much deeper insights than what they currently put forth.

1) I am not compelled by the artificial restriction to keep fossil production constant across the equity and cost-optimal scenarios. The authors' argument is given in the Methods ("*This is because we are interested in the impacts of redistribution, not the impact of tax on the absolute levels of production, which could reduce production in the absence of such constraints due to the higher costs of production arising from such a tax.*"). I do not find this reasoning to be compelling. The authors seem to want a 'ceteris paribus' comparison, but global costs are changing anyway from this reallocation compared to the cost-optimal case. As the authors acknowledge, relaxing this restriction will reduce global costs (since fossils overall become less attractive), and reduce the efficiency/equity trade-off that the authors emphasize in their conclusion. This global cost reduction will impact the relevant developing countries' finances, and therefore cannot be ignored. If, hypothetically, cheap fossil production in the US but not elsewhere is standing in the way of a faster transition to renewables, that's an important insight into how an energy transition would evolve. An examination of equity should be applied to as realistic a representation of such a transition as is possible in models.

2) The results depend on the production cost curves assumed for the different countries. The authors (in the SI only) cite one article by McGlade, and another unfinished PhD for updates, but offer no information on these cost parameters in the main text, nor any discussion about the uncertainty in future costs of production in literature. The IAM community must have their own set of cost curves – which in modeling protocols hopefully are accessible. The authors ought to include a discussion on and defend these costs (focusing on how they differ across fossil producers across the world), preferably representing the modeling community, not just their own assumptions, so that they can present more robust results with uncertainty.

3) There are many principles suggested in literature for how to allocate this scarce fossil production 'budget' – HDI is a good one, but not the only one. Another is GDP per capita, consistent with the Capacity principle, and consistent with the focus of this study on

financial flows. Another is historical responsibility, which could be implemented as an inverse of historical cumulative production shares. These additional scenarios would provide a richer representation of equity. The latter seem more appropriate choices for this piece, since HDI differs from monetary measures due to the health and education components, whose progress has little to do with fossil production other than through the financial flows that those generate.

4) Why look only at SSP2? I am guessing at least SSP1 and SSP5 will have a substantially different base of fossil production, and therefore different cost implications. SSP5 does not reach 1.5C but you can use other feasible stabilization scenarios.

Reviewer #4 (Remarks to the Author):

This paper discusses the consequences of redistribution of fossil production in the supply side to ensure the equitability between developed and developing countries. The topic of the paper is novel and it is relevant to the journal. The paper is well written and presented in a clear manner. The presented results show evident guidance for stakeholders and policymakers.

The model used in this paper is called "TIAM-UCL", which is an integrated assessment model (IAM) based on TIMES. To better understand this model for non-experts in IAM, a brief description of the model would be needed, e.g., mathematic equations of the objective function and constraints. In addition, cost assumptions are crucial to the cost optimisation carried out in this paper, hence a cost assumptions table for key technologies is required.

Two climate ambitions are investigated in this paper: 1.75 and 2 degrees of temperature increment. The corresponding carbon budgets are taken from the SR15 report. Unfortunately, a better-recognised and more interesting scenario of 1.5 is infeasible due to several factors, which compromises the significance of the conclusions.

The mechanism for redistribution is by adding different levels of carbon taxes on production. However, the choices of carbon taxes for VHHD, HHD and LMHD seem a bit random, where those numbers (1.1^4 , 0.9^4 , 0.6, 0.1) need to be clarified.

Response to reviewers

Manuscript: NCOMMS-19-36538-T, An equitable redistribution of unburnable carbon.

Reviewer comment	Authors' response
Reviewer 3	
This article examines the distributive implications of fossil-producing countries sharing the depleting market for fossils based on economic efficiency or equity in stringent climate stabilization scenarios. Past literature on equity has been dominated by mitigation cost sharing, until recently. The authors correctly point out that there has been growing attention to the issue of stranded fossil assets, including for developing economies that rely on fossils for supply and export. This article would be the first to quantitatively examine the financial implications of equity frameworks in distributing future fossil production. In this sense, the article is welcome.	We thank the reviewer for the constructive comments, which we have responded to in full below.
However, this article picks up on only one aspect of an equitable view of fossil extraction (how to allocate the remaining fossil production 'budget' – there are many others, as pointed out by the authors in their introduction (e.g. employment, diversification away from fossil dependence, stranded costs, etc). For a high-impact journal, the authors need to do a far more thorough analysis of this one aspect if they are to provide robust insights for policy. The authors choose a very limited set of scenarios from a much wider range of possibilities (as described below). A deeper exploration would potentially provide much deeper insights than what they currently put forth.	In this paper, we continue to focus on the specific equity criteria concerning level of development (HDI), as our research is primarily motivated by the debate focused on the rights of developing countries to exploit their domestic resources and the responsibility of countries with higher capacity to move quicker in respect of mitigation. This is also the perspective we bring to the paper in section 'pathways to managed decline'. To broaden the analysis undertaken in the paper, we have undertaken an alternative equity-based redistribution, based on the concept of accrued benefits from fossil fuel production. This provides an approach for comparison to the main HDI approach, to underline the very different impact on regions based on the use of different criteria. We have also looked at a lower demand case, using SSP1 demands (again using the HDI approach), to highlight the impact of smaller future fossil fuel production markets.
1) I am not compelled by the artificial restriction to keep fossil production constant across the equity and cost-optimal scenarios. The authors' argument is given in the Methods ("This is because	We have revisited this point, and decided that it is probably unnecessary to hold production levels at that observed under the optimal case – and important to see the reduction in production associated with the disincentives from the

we are interested in the impacts of redistribution, not the impact of tax on the absolute levels of production, which could reduce production in the absence of such constraints due to the higher costs of production arising from such a tax.”). I do not find this reasoning to be compelling. The authors seem to want a ‘ceteris paribus’ comparison, but global costs are changing anyway from this reallocation compared to the cost-optimal case. As the authors acknowledge, relaxing this restriction will reduce global costs (since fossils overall become less attractive), and reduce the efficiency/equity trade-off that the authors emphasize in their conclusion. This global cost reduction will impact the relevant developing countries’ finances, and therefore cannot be ignored. If, hypothetically, cheap fossil production in the US but not elsewhere is standing in the way of a faster transition to renewables, that’s an important insight into how an energy transition would evolve. An examination of equity should be applied to as realistic a representation of such a transition as is possible in models.	production tax. The model has subsequently been re-run to allow for this dynamic to be captured, with resulting reductions in production levels. The result is lower levels of production than observed in the optimal case, as would be expected. However, the increase in costs (relative to the optimal case) still holds in respect of our conclusions, due to the production of higher cost resources than would have otherwise been utilised. We also continue to conclude that developing countries do not get the expected benefit, due to the costs of production offsetting revenue gains for exports – and the ever decreasing markets in which to sell.
2) The results depend on the production cost curves assumed for the different countries. The authors (in the SI only) cite one article by McGlade, and another unfinished PhD for updates, but offer no information on these cost parameters in the main text, nor any discussion about the uncertainty in future costs of production in literature. The IAM community must have their own set of cost curves – which in modeling protocols hopefully are accessible. The authors ought to include a discussion on and defend these costs (focusing on how they differ across fossil producers across the world), preferably representing the modeling community, not just their own assumptions, so that they can present more robust results with uncertainty.	The fossil fuel supply curves used in TIAM have been subject to significant review and revision over the last 5-6 years. In fact, a core focus of the TIAM analysis has related to the role of fossil fuels, with many different publications.  • C. McGlade, P. Ekins, Un-burnable oil: An examination of oil resource utilisation in a decarbonised energy system, Energy Policy. 64 (2014) 102–112. doi:10.1016/j.enpol.2013.09.042. • C. McGlade, P. Ekins, The geographical distribution of fossil fuels unused when limiting global warming to 2 °C, Nature. 517 (2015) 187–190. doi:10.1038/nature14016. • C.E. McGlade, M. Bradshaw, G. Anandarajah, J. Watson, P. Ekins, A bridge to a low carbon future? Modelling the long-term global potential of natural gas, UKERC Research Report, London, UK, 2014. • N. Bauer, C. McGlade, J. Hilaire, P. Ekins, Divestment prevails over the green paradox when anticipating strong future climate policies, Nat. Clim. Chang. (2018). doi:10.1038/s41558-017-0053-1. • S. Bradley, G. Lahn, S. Pye, Carbon Risk and Resilience: How Energy Transition is Changing the Prospects for Developing Countries with Fossil Fuels, London, UK, 2018.

	https://www.chathamhouse.org/publication/carbon-risk-resilience-how-energy-transition-changing-prospects-countries-fossil.  • S. Pye, C. McGlade, C. Bataille, G. Anandarajah, A. Denis-Ryan, V. Potashnikov, Exploring national decarbonization pathways and global energy trade flows: a multi-scale analysis, Clim. Policy. 16 (2016) 1–18. doi:10.1080/14693062.2016.1179619. To address the shortcoming outlined by the reviewer, we have included a detailed section in the SI on the representation of fossil fuels in TIAM (SI section 5), including the development of resource supply curves. This has also been cross-referred to in the Methods section.
3) There are many principles suggested in literature for how to allocate this scarce fossil production ‘budget’ – HDI is a good one, but not the only one. Another is GDP per capita, consistent with the Capacity principle, and consistent with the focus of this study on financial flows. Another is historical responsibility, which could be implemented as an inverse of historical cumulative production shares. These additional scenarios would provide a richer representation of equity. The latter seem more appropriate choices for this piece, since HDI differs from monetary measures due to the health and education components, whose progress has little to do with fossil production other than through the financial flows that those generate.	It is true that there are different approaches one could use to allocate fossil fuels, in addition to the HDI. We chose to use the HDI criteria as we believe that it is a more rounded development indicator than GDP per capita. We are interested in the issue of development as the debate we are feeding into concerns discussions of fossil extraction and equity in the context of supporting development needs, notably the issue of revenue gains. Here we are teasing out the implications of this idea using a modelling approach. Therefore, we have provided a stronger justification for HDI versus other approaches in the Methods section. However, to develop the paper further, we have also implemented an ‘accrued benefits’ approach, which consider accumulated rents over time. The different regional groupings are then modelled using the same redistributive mechanism – the carbon-based production tax. We believe that including this additional basis for equitable redistribution, described fully in SI section 3 but also referred to in the paper, provides a useful comparison to the HDI approach – and highlights the considerable differences from choosing an alternative method.
4) Why look only at SSP2? I am guessing at least SSP1 and SSP5 will have a substantially different base of fossil production, and therefore different cost implications. SSP5 does not reach 1.5C but you can use other feasible stabilization scenarios	We thank the reviewer for the suggestion, and have expanded the paper by including scenarios based on SSP1 demand drivers. The approach and results are described in SI section 2, and referred to in the main paper.

Reviewer 4	
This paper discusses the consequences of redistribution of fossil production in the supply side to ensure the equitability between developed and developing countries. The topic of the paper is novel and it is relevant to the journal. The paper is well written and presented in a clear manner. The presented results show evident guidance for stakeholders and policymakers.	We thank the reviewer for the constructive comments, which we have responded to in full below – and for recognising the value and relevance of the paper.
The model used in this paper is called "TIAM-UCL", which is an integrated assessment model (IAM) based on TIMES. To better understand this model for non-experts in IAM, a brief description of the model would be needed, e.g., mathematic equations of the objective function and constraints. In addition, cost assumptions are crucial to the cost optimisation carried out in this paper, hence a cost assumptions table for key technologies is required.	We have added SI section 4 to the paper which provides a brief primer on TIMES and describes the key equation and constraints. We have also further developed the assumptions table on key assumptions in SI section 6, to include the cost assumptions for the key technology groups. In addition, in response to Reviewer 3, we have added a detailed SI section 5 on the fossil resource supply sector in TIAM-UCL.
Two climate ambitions are investigated in this paper: 1.75 and 2 degrees of temperature increment. The corresponding carbon budgets are taken from the SR15 report. Unfortunately, a better-recognised and more interesting scenario of 1.5 is infeasible due to several factors, which compromises the significance of the conclusions.	While there has been significant discussion of 1.5C pathways notably under the IPCC special report, as the reviewer notes, TIAM-UCL does not solve for this in a SSP2 world. We could of course find ways for the model to solve but this would mean using assumptions that push the boundaries of what we deem feasible, e.g. by increasing biomass availability beyond sustainable limits. For example, bioenergy global resource used in many of the IAMs solving for 1.5C is double what we use here, and results in a much stronger role for BECCS. It is also worth noting that two of the main IAMs (IMAGE and WITCH) also do not solve for this target under SSP2 demand (as reported in Rogelj, J., Popp, A., Calvin, K. V., Luderer, G., Emmerling, J., Gernaat, D., ... & Krey, V. (2018). Scenarios towards limiting global mean temperature increase below 1.5 C. Nature Climate Change, 8(4), 325.) In addition to noting the uncertainty for future fossil demand in the paper due to the role of CCS/BECCS, we now also reflect on the implication of more stringent climate targets e.g. under 1.5C.

The mechanism for redistribution is by adding different levels of carbon taxes on production. However, the choices of carbon taxes for VHHD, HHD and LMHD seem a bit random, where those numbers (1.1^4, 0.9^4, 0.6, 0.1) need to be clarified.	We have simplified our approach to selecting carbon-based production prices used in the model, as described in the Methods section. Starting tax levels in 2020 represent the range of prices seen today (either explicit or implicit). We then apply a higher and lower annual growth rate, establishing two tax variant trajectories for the VHHD group; the HHD group incurs a tax at 50% of the VHHD level, while the LMHD group sees no tax at all.
--	---

REVIEWER COMMENTS

Reviewer #3 (Remarks to the Author):

I am satisfied that the authors have made substantial and beneficial improvements to address my comments. I think the historical benefits scenario, interpreted as rents from fossil production, is consistent with the supply-side focus of the paper, and provides a useful alternative to HDI-based allocation. I also think the addition of SSP1 is useful, as a lower demand scenario (SSP3 probably doesn't yield either 1.75C or 2C, so it is superfluous).

However, one remaining suggestion would be very helpful in my eyes for readers (in light of the additional scenarios):

I see no descriptive characterization, either in the manuscript, or SI, of the relationship between HDI and fossil production. That is, measures of well-being are causally related to energy consumption, not production. Given that this paper focus on production gains, it would be beneficial to characterize the extent to which HDI correlates to fossil production - many of the key fossil producers (e.g. Russia, middle east) are not low HDI, and many low HDI countries are importers. Providing some descriptive information on countries mapping to these indicators can be an very informative to illuminates the two contrasting scenarios - one based on HDI and the other based on fossil rents.

I have reviewed the article and the responses to Reviewer 4's comments. I do think that the authors have provided the requested clarity on the assumptions behind the chosen carbon taxes, but I still find the choices arbitrary - in particular, the 50% of VHHD for the HHD. Why not peg the decrease to the (wtd) avg GDP of the group (and if this is approximately the basis, then it should be stated), or some other empirical basis?

Response to reviewers

Manuscript: NCOMMS-19-36538A, An equitable redistribution of unburnable carbon.

Reviewer comment	Authors' response
Reviewer 3	
I am satisfied that the authors have made substantial and beneficial improvements to address my comments. I think the historical benefits scenario, interpreted as rents from fossil production, is consistent with the supply-side focus of the paper, and provides a useful alternative to HDI-based allocation. I also think the addition of SSP1 is useful, as a lower demand scenario (SSP3 probably doesn't yield either 1.75C or 2C, so it is superfluous).	We thank the reviewer for the comments on the last paper version, which we believe have significantly improved the paper.
However, one remaining suggestion would be very helpful in my eyes for readers (in light of the additional scenarios): I see no descriptive characterization, either in the manuscript, or SI, of the relationship between HDI and fossil production. That is, measures of well-being are causally related to energy consumption, not production. Given that this paper focus on production gains, it would be beneficial to characterize the extent to which HDI correlates to fossil production - many of the key fossil producers (e.g. Russia, middle east) are not low HDI, and many low HDI countries are importers. Providing some descriptive information on countries mapping to these indicators can be very informative to illuminates the two contrasting scenarios - one based on HDI and the other based on fossil rents.	Thank you for highlighting this point. As the reviewer rightly points out, development progress is much more closely correlated to energy consumption whereas the link between development and fossil production is entirely dependent on a range of factors, including the quality of governance, and the translation of centralised economic rents into broad-based development. The failure to do so is one of the well-known impacts of the resource curse. We have now added SI section 7 to provide a more comprehensive discussion of some of these issues. We start by explaining the selection of HDI as a metric for exploring equity-based allocation of production. This is because much of the literature on equity considerations around future fossil extraction has been grounded in the idea of 'common but differentiated responsibilities and respective capabilities'. An important theme within discussions around differentiated responsibilities and respective capabilities has been the 'need to develop', as something which differentiates countries, thereby placing different responsibilities on them, in terms of how urgently they should decarbonise, or forgo fossil fuel development. Least developed countries would be considered to have greater need to delay decarbonisation, or continue to invest in fossil fuels, than the most developed countries, according to this line of argument. We highlight that while the HDI metric is a means of identifying the development status of a country, it is true that there is a weak correlation between development status and fossil fuel production, or availability of fossil fuel reserves. As the reviewer suggests, this would have two particularly

	important implications for the implementation of an HDI-based redistribution mechanism. First, as not all low-HDI countries are producers, it would not benefit all low-HDI countries, and could disadvantage some who were importers (because of higher costs of fuels). Second, because clearly not all countries with fossil fuels are high HDI, there is not a clear connection between having fossil fuel reserves and having equitable development – the well-known ‘resource curse’. It is interesting to note that Africa is in the low HDI group but in the middle group under the accrued benefits criteria. As noted, these important issues are now discussed in the SI 7. This section goes into some more detail as to the reasoning behind the use of the different criteria, and develops in more detail the two key points, summarised above, that follow from the weak correlation between fossil fuel reserves and level of development. As part of that SI we have also reflected on the current production levels of fossil fuels across the different countries (and regions used in the model). In addition, these issues are also further emphasized in the main text within the section ‘Re-considering production-based equity perspectives’. The existing text also refers to the fact that LMHD countries that are fossil fuel importers could be disadvantaged by redistributive measures that increased costs, at the end of the opening section of the paper.
I have reviewed the article and the responses to Reviewer 4’s comments. I do think that the authors have provided the requested clarity on the assumptions behind the chosen carbon taxes, but I still find the choices arbitrary – in particular, the 50% of VHHD for the HHD. Why not peg the decrease to the (wtd) avg GDP of the group (and if this is approximately the basis, then it should be stated), or some other empirical basis?	The approach taken here was to half the tax rate on the basis of Group 1 (HHD) HDI score being in the middle of the range provided by Group 2 and Group 0. The following text (in red) has been added in the Methods section to clarify this – “For Group 1 countries, the tax trajectories (orange trend lines) are at 50% of the level for Group 2. This level was determined based on the distribution of population-weighted HDI scores across the three groups, where Group 1’s score of 0.74 was in the middle of the range between Group 2 (0.89) and 0 (0.58). Group 0 countries do not incur any carbon tax on production. Both equity cases followed the same basis for comparability.”

REVIEWERS' COMMENTS:

Reviewer #3 (Remarks to the Author):

In response to my request to contrast HDI- vs production-based status of countries, the authors have provided a thoughtful and important section in the manuscript and SI that places production-based equity perspectives and its limitations in the broader landscape of equity frameworks - I am grateful to the authors for taking the comments seriously. I am satisfied that the article is suitable for publication and makes a valuable contribution.